# OpenReview forum: "Virus Infection Attack on LLMs: Your Poisoning Can Spread "VIA" Synthetic Data"
_NeurIPS.cc/2025/Conference — NeurIPS 2025 spotlight_

### Official Review · Reviewer_6brq · 2025-06-08

**Clarity:** 4
**Significance:** 3
**Originality:** 4
**Rating:** 5
**Confidence:** 4

**Summary:**

In this work, the authors conduct a study on whether backdoors in LLMs will be passed on to downstream LLMs through a standard process of distillation. The findings show that it is unlikely given the low probability of triggering such backdoors using standard datasets. The authors then propose an approach to conduct training-time poisoning more effectively such that the backdoors are more likely to pass on. The proposed approach is evaluated using three different backdoors.

**Questions:**

Can you show whether similar results can be obtained on other LLMs?

**Ethical Concerns:**

["NO or VERY MINOR ethics concerns only"]

**Final Justification:**

I remain positive about this work after the rebuttal.

**Limitations:**

Although there are no explicit discussions on the limitations, the approach has been clearly presented and all limitations are clearly implied.

**Paper Formatting Concerns:**

Page 2, the paragraph starting from line 51.

Comment: It is known that knowledge distillation can be applied to weaken attacks such as data-poisoning based backdoor attacks. I would suggest that the authors should at least add a discussion on such existing results.

Page 4: "... suggesting that the proportion of poisoning content in synthetic data is significantly lower
118 than that in the training corpus of the upstream model."

Comment: This is rather interesting. I would be curious to know why this would be the case. I would suggest to contrast the distribution of the original poisoning samples, against this one, and try to form some conjecture why the learning would lead to such a distribution with more empirical study - such as making the same contrast with a data poisoning attacker that is stealthier.

Page 6: "Shell Construction (SC)"

Comment: The design of the shell still seem simplistic

**Quality:**

3

**Strengths And Weaknesses:**

Strength

The topic of the study is interesting. From what I can tell, this is a novel approach.

The paper is well-presented. While the first part, i.e., the part on RQ1, yields expected empirical results, the second part, i.e., the design of the more persisting training-time poisoning is interesting.

The experimental results show effectiveness of the proposed approach.

Weakness

The part on the HPS score, while reasonable and intuitive, can be better studied and argued. For instance, it would be interesting to know whether such strategy is transferrable - for instance, if the attacker can only see part of the training dataset, whether he/she can still obtain effective HPS.

The experimental evaluation can be strengthened with more models - as of now it is only one model.

---

> ### Author Rebuttal · Authors · 2025-07-30
>
> We sincerely appreciate your commendation on our work!
>
> Following your suggestions, we will expand the discussion on (i) leveraging knowledge distillation to counter backdoor poisoning, and (ii) visualizing the distribution differences between clean and poisoned training samples.
>
> In addition, to address your concern regarding the transferability of the HPS score at the distribution level and the influence of LLMs, we have conducted two supplementary experiments, detailed below.
>
> ## Is HPS Transferrable?
>
> Though HPS is grounded in a theoretical lower bound derived from our objective, it remains an empirical score in practice. Computing HPS on only a subset of the training data or on an entirely different dataset with a different distribution may therefore influence its reliability.
> However, just as the hijacking words demonstrated in our paper, we believe these words can be extracted from various SFT datasets or a subset.
>
> We provide an experimental analysis of this issue for your reference.
>
> | Dataset Length                              | Spearman Correlation | Jaccard Similarity |
> |---------------------------------------------|:----------------------:|:--------------------:|
> | 25000 vs. 25000  (Tulu3-SFT)                | 1.0                  | 1.0                |
> | 15000  vs. 25000  (Tulu3-SFT)               | 1.0                  | 1.0                |
> | 10000  vs. 25000 (Tulu3-SFT)                | 0.9976               | 1.0                |
> | 5000  vs. 25000   (Tulu3-SFT)               | 0.9947               | 1.0                |
> | 10000 on OpenO1-SFT vs. 25000 on Tulu-3-SFT | 0.9528               | 0.1165             |
>
> These results indicate that HPS rankings are robust to sample size and even moderately robust to a data distribution which is totally different, supporting its practical transferability.
>
> ## Results obtained on other LLMs
>
> We repeated the data-poisoning experiments on two LLMs, Qwen2.5-7B and Llama3-8B.
>
> | Backbone LLM | Method | Upstream Model ASR | Synthetic Data IR |
> |--------------|---------------|:--------------------:|:-------------------:|
> | Qwen2.5-7B   | HPS       | 53.65              | 50.6              |
> | Qwen2.5-7B   | HPS-SS    | 41.46              | 29.8              |
> | Qwen2.5-7B   | HPS-SS-SC | 82.92              | 12.0              |
> | Llama3-8B    | HPS       | 26.82              | 72.4              |
> | Llama3-8B    | HPS-SS    | 46.34              | 57.9              |
> | Llama3-8B    | HPS-SS-SC | 78.04              | 22.9              |
>
> This table indicates that VIA consistently raises the infection rate and produces non-trivial downstream ASR across backbones, confirming the general utility of our framework.
> Besides, Qwen2.5-7B is more susceptible to the initial poisoning (higher upstream ASR) but exhibits stronger resistance to propagation (lower IR) than Llama3-8B. Investigating the architectural or training factors behind this discrepancy seems an interesting direction for future work.

---

### Official Review · Reviewer_7ogA · 2025-06-30

**Clarity:** 3
**Significance:** 3
**Originality:** 3
**Rating:** 5
**Confidence:** 4

**Summary:**

The paper introduces a novel method that compels poisoned data to infect models trained on synthetic text produced by a compromised language model. First, the authors demonstrate that standard data-poisoning techniques fail to transfer to the downstream model. To overcome this limitation, they propose the **Virus Infection Attack (VIA)**, which raises the **Infection Rate (IR)**—the proportion of poisoned samples in the synthetic dataset.
Experiments show that VIA increases the IR, injecting more poisoned examples into the synthetic training corpus, while simultaneously lowering the attack-success rate (ASR) on the original language model.

**Questions:**

- Q1: In figure 2 why are the graphs not shown for smaller amount of poisoning? Why is for example the vanilla poisoning
- Q2: What about distributional shifts between the Poisoning Samples and the Query Samples? Is the assumption here that these two distributions are the same? It is not quite clear to me what dataset was used for the query set in the experiments.
- Q3: This relates to question 2: What is the ASR on a model trained on this synthetic dataset? As far as I understand, a high IR does not necessarily lead to a ASR on the trained model as this highly depends on the query set and on the other datasets that are used in addition to the synthetic dataset for the training.

**Ethical Concerns:**

["NO or VERY MINOR ethics concerns only"]

**Final Justification:**

My main concern was about the downstream ASR. However, the rebuttal has adequately addressed my concerns, which is why I am raising my score.

**Limitations:**

- The limitation that a high IR does not necessarily mean a high ASR on the trained model was discussed only very briefly. I think some experiments are needed here for evaluating that.

**Paper Formatting Concerns:**

Making so many values in Table 2 bold is a bit confusing. Maybe only highlight the best and second-best for each of the columns. This would greatly improve readability.

**Quality:**

3

**Strengths And Weaknesses:**

**Strengths:**
- The paper is well written
- The concept is easy to understand
- The concept of propagating poisoned data to the synthetic dataset is very interesting and timely

**Weaknesses:**
- The ASR on the original model is reduced. While this is not a problem in itself, it would be nice to successfully infect both the original and trained models.
- Even though there might be a high IR, this does not necessarily guarantee a high ASR on the downstream model trained using the synthetic data. An experiment evaluating the ASR on the LLM trained using the synthetic data is missing.

---

> ### Author Rebuttal · Authors · 2025-07-30
>
> We sincerely appreciate your constructive comments and valuable insights.
>
> After carefully reading, we understand that your primary concern is the drop in the upstream model’s ASR after applying VIA.
> Actually, the extent of this ASR decrease appears to have been overthought; therefore, below we clarify three points for your kind reference:
>
> + The drop is specific to backdoor poisoning attacks. For data-poisoning attacks (Sentiment Steering, Knowledge Injection, and Biased Recommendation), the upstream ASR is on par with or even higher than that of vanilla poisoning; therefore, the phenomenon is **not universal**.
> + Even for backdoor poisoning attacks, the drop can be easily mitigated by a simple **MIXUP**, as shown in our paper.
> + The ASR of downstream models poisoned with VIA is consistently higher than that achieved by traditional poisoning attacks.
>
> In summary, VIA enables the adversary to (i) retain competitive ASRs on upstream models (often after a lightweight mix-up step) and (ii) obtain significantly higher ASRs on downstream models IRs on synthetic data. We therefore view the “high IR but lower upstream ASR” as an intriguing observation rather than a limitation.
>
> Also, we'd like to answer the remaining questions to address your concerns.
>
> > Q1: In figure 2 why are the graphs not shown for smaller amount of poisoning?
>
> We omitted the lower PR results for vanilla poisoning in Figure 2 to keep the focus on VIA; nevertheless, we agree that showing these points would make the comparison more complete.
>
> Below we supply the missing values and will add the corresponding curves in the revised figure.
>
> | Poisoning Rate | Upstream Model ASR | Synthetic Data IR |
> |:----------------:|:--------------------:|:-------------------:|
> | 0.001          | 53.65              | 0.00              |
> | 0.005          | 90.24              | 0.00              |
>
> > Q2: What about distributional shifts between the Poisoning Samples and the Query Samples? Is the assumption here that these two distributions are the same? It is not quite clear to me what dataset was used for the query set in the experiments.
>
> Thank you for raising this important point.
>
> Our theoretical bound indeed assumes that the poisoning distribution and the query distribution are identical. This assumption is natural, as it aligns with standard analyses of synthetic-data pipelines. However, **the derivation does not imply that VIA fails when they are different; it simply shows that the infection rate will not reach its theoretical maximum under distribution shift.** Empirically, VIA still produces effective poisoning.
>
> To illustrate, if we inject an incorrect math knowledge (e.g., "e $\approx$ 3.14") via VIA, its impact might be negligible when downstream queries are unrelated to mathematics. However, once mathematical questions appear, even if they do not concern the value of e, the poisoned knowledge can surface, thereby yielding a significant higher ASR than traditional methods.
>
> In our experiments, queries and upstream training datasets are separately sampled from the same dataset. The three datasets includes Tulu-3-SFT (for general-purpose), OpenO1 (for reasoning), and Alpaca (for backdoor).
>
> > Q3: This relates to question 2: What is the ASR on a model trained on this synthetic dataset?
>
> The downstream ASR depends on the effective infection rate in the final synthetic dataset used for fine-tuning. Empirically, as long as the IR remains sufficiently high, the downstream ASR could never be lower than the upstream ASR. **In our experiments, the IR is high enough** to withstand at least a 50× dilution before the poisoning proportion becomes negligible. Consequently, once the downstream training corpus contains the same or higher poisoning (infection) rate than the upstream corpus, the downstream ASR equals or exceeds the upstream ASR.
>
> > More experiments from IR to ASR...
>
> In the Appendix, we already include experiments on data-poisoning attacks that address this issue. The additional backdoor results will be incorporated into the revised paper for completeness.
>
> Experiments on Backdoor Poisoning Attacks:
>
> | Backdoor Method | VIA Used? | Poisoning Rate | Upstream Model ASR | Infection Rate | Downstream Model ASR |
> |-----------------|-----------|:----------------:|:--------------------:|:----------------:|:----------------------:|
> | BadNet          | No        | 2%             | 99.5%              | 0.15%          | 1.0%                 |
> | BadNet          | with VIA  | 2%             | 56.5 %             | 52.97%         | 64.0%                |
> | CTBA            | No        | 2%             | 100.0 %            | 0.45%          | 2.5%                 |
> | CTBA            | with VIA  | 2%             | 18.5%              | 61.42%         | 30.5%                |
> | MTBA            | No        | 2%             | 95.5%              | 0.30%          | 2.5%                 |
> | MTBA            | with VIA  | 2%             | 64.0%              | 58.10%         | 71.5%                |
> | Sleeper         | No        | 2%             | 24.5%              | 0.00%          | 1.0%                 |
> | Sleeper         | with VIA  | 2%             | 54.0%              | 65.72%         | 67.5%                |
> | VPI             | No        | 2%             | 98.0%              | 0.02%          | 0.0%                 |
> | VPI             | with VIA  | 2%             | 52.0%              | 63.22%         | 62.5%                |
>
> The results are consistent with our analysis in the paper.

---

> > ### Comment · Reviewer_7ogA · 2025-08-04
> >
> > Thank you very much for your detailed rebuttal.
> > All my conerns have been addressed, especially the one about the downstream ASR. Therefore, I will raise my score.

---

> > > ### Author Response · Authors · 2025-08-04
> > >
> > > Thank you very much for your timely response! We're so glad we addressed your concerns and sincerely appreciate the increased score :)

---

### Official Review · Reviewer_Xcj8 · 2025-07-02

**Clarity:** 1
**Significance:** 2
**Originality:** 2
**Rating:** 4
**Confidence:** 3

**Summary:**

The paper investigates the threat model of propogation of data poisoning through synthetic data. In other words, the attacker is able to poison a dataset \(\tilde {\cal D}\) and a developer trains a model A on that dataset. The model A is then used to generate synthetic data for another model B. The paper studies the question of how \(\tilde {\cal D}\) can be modified such that the model B is controllable (has specific biases, has secret behavior with a passcode, etc)

The authors identify why traditional data poisoning on \(\tilde {\cal D}\) does not result in B being poisoned, namely that the synthetic data generated by A is very varied while the poisoning attack is narrow. An attack which causes model A to have fake knowledge of politician X is not transferred to B if the synthetic data never mentions politician X.
To improve this transfer, the authors design a method where the first poisoning is designed to make model A mention e.g. politician X in a broad range of situations.

The attack is evaluated in a broad range of situations and shown to lead to good transfer to model B.

**Questions:**

1. Can you clarify what is shown in the various graphs and tables? Namely, do any of them show the ASR on the downstream model, i.e. the model which is trained on the synthetic data which is generated from the model which has been poisoned?
2. Can you give a more detailed analysis of the stealthiness of VIA, particularly as you vary the poisoning rate on the original model?
3. Can you elaborate on the relevance of the threat model and why it i a realistic case to consider?

**Ethical Concerns:**

["NO or VERY MINOR ethics concerns only"]

**Final Justification:**

This paper poses an interesting threat model to the common practice of synthetic data training.
The paper is technically solid. Initially the authors' results didn't really seem to support their main thesis, as they focussed on the attack success rate on the intermediate model, which isn't relevant for their approach. However, they have updated their results to showcase the results which reinforce the focus of the paper, so I now believe the paper is a decent contribution.

**Limitations:**

Yes, mostly

**Quality:**

3

**Strengths And Weaknesses:**

### Strengths
+ The paper identifies an interesting question that is only becoming more relevant as the use of synthetic data is scaling up.
+ The attack is straightforward and makes sense intuitively

### Weaknesses
+ The paper is unclear in several places and could benefit from some editing. In particular, figure 2 (the figure with the main/interesting results!) is very small and the labels are tiny and confusing--it is hard to tell whether it is supposed to read as $P_{\tilde \theta}$'s ASR (i.e. with an apostrophe)or $P_{\tilde \theta'}s$ (i.e. with a $\prime$). I am assuming that the label is a typo and it should read $P_{\theta'}'s$ (i.e. the ASR of the method on model *B* after poisoning the dataset for model A's training and using the synthetic data from model A to train model B). However, after trying to understand a bit further, I believe that figure 2 might *not* show the ASR of $P_{\theta '}$. In fact, I can only find this information in figure 7 in the appendix, which shows that the ASR of the VIA attack remains high for model B. I am confused why this graph doesn't appear in the main text, given that none of the other results in the paper seem to actually target the key contribution of allowing data poisoning to have high ASR even when propogated through intermediate models. I'm also confused why table 1 and 2 focus so much on the ASR for the poisoned upstream model--from the provided motivation, this doesn't really matter for the problem setting.

+ Although the authors address the stealthiness of the attack in section 4.3, it is quite brief, and the perplexity-based filtering is quite a simple method. Looking at the samples, they appear very blatant and I would suspect that a small LLM could be fine-tuned to detect these data poisoning statements. I would be curious whether this is possible, or if it would require a larger LLM to detect these abnormalities (which may add too much overhead). In general, the paper could do with a more detailed study of the trade-off between the downstream model ASR and the stealthiness of the attack. For instance, we could assume a 0.1% or 1% FPR and see what ASR is obtainable with a TPR of 0%, 1%, 5%, 20%, etc.

+ More generally, the importance of the threat model is somewhat unclear. It is generally true that data poisoning is an important issue. However, is this setup particularly different in the sense of necessary safeguards relative to the traditional data poisoning setup with public data? It would seem that the traditional defenses to detect data poisoning would also apply to the synthetic data. Perhaps there is a mistaken belief in the community that synthetic data is necessarily safe and doesn't need to be filtered or checked in any way. However, I am not aware of such a belief.

---

> ### Author Rebuttal · Authors · 2025-07-30
>
> We sincerely appreciate your thoughtful feedback and critical review of our study.
> Below, we address your specific concerns, and welcome any further follow-up questions.
>
> # Clarification of Misunderstandings
> ## Figure 2
> > "...it is hard to tell whether it is supposed to read as $P_{\tilde{\theta}}$'s ASR or $P_{\tilde{\theta}''}$. I am assuming that the label is a typo and it should read $P_{\theta'}'.$".
>
> In Figure 2's caption, we have mentioned and bolded that Figure 2 focuses on the "Performance Comparison of Poisoned **Upstream Model's ASR**", which represents the ASR of $P_{\tilde{\theta}}$ instead of a typo. We will resize the labels in this figure and revise the notations following your feedback.
>
> ## Threat Model
>
> > More generally, the importance of the threat model is somewhat unclear. It is generally true that data poisoning is an important issue. However, is this setup particularly different in the sense of necessary safeguards relative to the traditional data poisoning setup with public data?
> > ...Although the authors address the stealthiness of the attack in section 4.3, it is quite brief, and the perplexity-based filtering is quite a simple method....
>
> We respectfully clarify some key points:
> - VIA is not a new method designed for synthetic data scenarios but a propagation-enabling framework (a plugin) for current attacks. Thus, *the core question here is not whether VIA itself can be defended against (as it operates jointly with other attacks), but whether it introduces* ​​**additional** *exposure risks​​ beyond conventional poisoning (Section 4.3, Line 282), which is also the target for our stealthiness analysis*. For instance, if an already unstealthy attack (A1) remains detectable after VIA reformatting, this is acceptable. If a stealthy attack (A2) becomes detectable post-VIA, this would be a limitation.
> - Before our work, the security of training on synthetic data had not been studied. Consequently, the threat posed by VIA remained unexposed. As a result, (i) practitioners were unaware of this specific risk and, at best, could rely only on generic poisoning defenses; (ii) no adaptive countermeasures against VIA existed.
> - To the best of our knowledge, neither prior research nor engineering practice has addressed the filtering of backdoor or data-poisoning attacks in synthetic data.
> - Our proposed framework (VIA) fundamentally differs from previous attacks, as it inject poisoning content *within* a clean sample rather than into training datasets directly. This represents a new poisoning framework distinct from conventional approaches.
> - The defenses we propose are tailored specifically to VIA and constitute the first adaptive defense against this threat under realistic settings.
> - In next parts, we further evaluate the stealthiness of VIA based on your feedback for your kind reference. However, even when training a classifier with **full knowledge** of poisoning content and strategies, our experiments show detectors fail to reliably distinguish poisoned samples.
>
> # Supplemental Experiments
>
> In this part, we provide additional experiments to address your concerns regarding defenses.
>
> ## Stealthiness & Defenses
>
> ### Fine-tuning an LLM for Detection
>
> Based on your suggestion, we trained a classifier to discriminate clean and poisoned samples in sentiment steering experiments. We observe that this strategy can detect simple VIA-based poisoning (e.g., injections at the beginning). However, the detection accuracy degrades significantly when poisoning content is wrapped with our SC strategy, approaching randomly guessing. Therefore, such a strategy seems to underperform our proposed defenses.
>
> | Method        | Pre.  | Rec.  | Acc.  | F1    |
> |---------------|-------|-------|-------|-------|
> | VIA+Start     | 99.41 | 96.04 | 97.71 | 97.70 |
> | VIA+HPS       | 76.51 | 56.42 | 68.85 | 64.95 |
> | VIA+HPS+SC    | 56.84 | 60.33 | 56.28 | 58.53 |
> | VIA+HPS+SS    | 63.82 | 51.72 | 61.42 | 57.14 |
> | VIA+HPS+SS+SC | 50.00 | 69.27 | 48.85 | 58.07 |
>
> When compared with our defense:
> - This detection strategy demonstrates limited effectiveness compared to our proposed defenses.
> - These results corroborate the analysis and conclusions (i.e., SC improves stealthiness) presented in Section 4.3 of our paper.
>
> Moreover, we respectfully note that such a defense strategy, i.e., training an LLM to detect poisoned samples, may have limited practical applicability. This approach requires unrealistically strong threat model assumptions, specifically that the defender has prior knowledge of both the exact poisoning content and its data distribution. In contrast, our proposed adaptive defense against VIA requires only knowledge of the poisoning mechanism, making it both more practical and widely applicable.
>
> > Although the authors address the stealthiness of the attack in section 4.3, it is quite brief, and the perplexity-based filtering is quite a simple method.
>
> While we fully understand your valid concern regarding the need for more effective defense, we respectfully argue that **designing a more powerful defense indeed exceeds the workload of an academic article**.
> Besides of proposing a new defense, this paper has already 1) explored an overlooked yet highly relevant poisoning scenario, ii) conducted a systematic analysis of the resilience and the limitation of previous attacks, and iii) developed a novel poisoning framework to reveal such a threat.
> While we agree that developing comprehensive defenses would be valuable, we believe such work would constitute a separate research endeavor beyond the scope of this paper.
>
> ### Tradeoff between Stealthiness and Downstream ASR
>
> Following your feedback, we formalize the computation of IR after filtering. Given the Infection Rate (IR) of the synthetic dataset, and the FPR and TPR of the poisoning detector, the IR after filtering can be computed by
>
> $IR After Filtering=FN/(FN+TN)=\frac{(1-TPR)\cdot IR}{(1-TPR)\cdot IR+(1-FPR)\cdot (1-IR)}$.
>
> Therefore, based on results of the above detector, the IR after filtering could be shown as:
>
> | Method        |    IR    | FPR on Detection | TPR on Detection | IR After Filtering |
> |---------------|--------|:------------------:|:------------------:|:--------------------:|
> | VIA:HPS+SS    | 57.92% | 0.02             | 0.944            | $\approx$ 0%       |
> | VIA:HPS+SS+SC | 22.98% | 0.02             | 0.216            | 19%                |
>
> A 19% IR demonstrates that the downstream model will exhibit a higher ASR compared with its upstream model, indicating that more powerful defenses are required to improve the TPR for detection.
>
> The downstream model ASR results further validate these findings.
>
> ## Downstream Model ASR
>
> >  In fact, I can only find this information in figure 7 in the appendix, which shows that the ASR of the VIA attack remains high for model B. I am confused why this graph doesn't appear in the main text, given that none of the other results in the paper seem to actually target the key contribution of allowing data poisoning to have high ASR even when propogated through intermediate models.
>
> We agree with you that it is essential to exhibit the experimental results for the downstream model's ASR when trained with VIA. We will move Figure 7 and its corresponding experiments to the main text.
>
> We also add the downstream model ASR results for all methods in Tables 1 and 2 below. These results will be incorporated as new columns in both tables.
>
> Experiments on Data Poisoning Attacks:
>
> | Method            | Task                  | Upstream Model ASR | Downstream Model ASR |
> |-------------------|-----------------------|:--------------------:|:----------------------:|
> | Vanilla Poisoning | Sentiment Steering    | 100.00             | 0.20                 |
> | with VIA          | Sentiment Steering    | 78.04              | 78.18                |
> | Vanilla Poisoning | Knowledge Injection   | 100.00             | 0.22                 |
> | with VIA          | Knowledge Injection   | 100.00             | 100.00               |
> | Vanilla Poisoning | Biased Recommendation | 5.26               | 0.06                 |
> | with VIA          | Biased Recommendation | 84.31              | 78.94                |
>
> Experiments on Backdoor Poisoning Attacks:
>
> | Backdoor Method | VIA Used? | Upstream Model ASR | Downstream Model ASR |
> |-----------------|-----------|:--------------------:|:----------------------:|
> | BadNet          | No        | 99.5               | 1.0                  |
> | BadNet          | with VIA  | 56.5               | 64.0                 |
> | CTBA            | No        | 100.0              | 2.5                  |
> | CTBA            | with VIA  | 18.5               | 30.5                 |
> | MTBA            | No        | 95.5               | 2.5                  |
> | MTBA            | with VIA  | 64.0               | 71.5                 |
> | Sleeper         | No        | 24.5               | 1.0                  |
> | Sleeper         | with VIA  | 54.0               | 67.5                 |
> | VPI             | No        | 98.0               | 0.0                  |
> | VPI             | with VIA  | 52.0               | 62.5                 |
>
> The results are consistent with our conclusions in the paper.
>
> Regarding the low ASR observed in upstream models for VIA-enhanced backdoor attacks, we have provided a detailed analysis in Section 4 and proposed VIA (mixup) as an effective solution to address this limitation.
>
> >  I'm also confused why table 1 and 2 focus so much on the ASR for the poisoned upstream model--from the provided motivation, this doesn't really matter for the problem setting.
>
> While ​​the​​ upstream model's ASR ​​is not relevant​​ to ​​the propagation​​ of poisoning, we respectfully ​​maintain​​ that it is necessary to include this part of ​​the​​ experiments. It measures whether ​​the​​ upstream model's ASR ​​is reduced​​ after using VIA, which is important for investigating potential drawbacks of our framework.

---

> > ### Author Response · Authors · 2025-08-06
> >
> > Dear Reviewer Xcj8,
> >
> > Thank you again for your valuable time and detailed review. We have posted a rebuttal where we sought to address your concerns with new empirical evidence and further clarifications.
> >
> > We would be grateful to know if our response has helped clarify these points and would be happy to discuss any remaining questions you may have.
> >
> > Thank you once more for your insights.
> >
> > Best regards,

---

> > > ### Comment · Reviewer_Xcj8 · 2025-08-06
> > >
> > > Thanks for the detailed rebuttal, and apologies for the delayed response.
> > > I have looked over the rebuttal and I welcome a lot of the changes that you say you will make in the paper. I think they will make the paper more focussed on testing the key hypotheses that you pose in the paper.
> > >
> > > The new columns for tables 1 and 2 make it much more obvious that VIA is an effective way to propogate attacks through the intermediate model and achieve much higher ASR compared to the vanilla approach.
> > >
> > > I am still a bit confused about the stealthiness of the attack--the transcripts in the paper seem very blatant, and my understanding of the capabilities of modern LLMs would lead me to guess they can detect these given a prompt such as (an elaboration on) 'monitor for irregularities'. However, I agree that your paper cannot be expected to cover everything. Therefore, I will raise my score--thank you again for the rebuttal, which has definitely changed my mind on the merits of the paper.

---

> > > > ### Author Response · Authors · 2025-08-06
> > > >
> > > > Thank you very much for your timely response and for raising the score! We appreciate your feedback and are glad to hear that our revisions have addressed most of your initial concerns.
> > > > As promised, we will revise Figure 2 for better clarity and include downstream ASR results to further demonstrate VIA’s effectiveness.
> > > >
> > > > Regarding the stealthiness of the attacks, we’d like to clarify that the examples in the paper are intentionally brief fragments (typically under 100 tokens) to clearly illustrate the poisoning mechanism. In our experiments, they are embedded within much longer texts, making manual detection labor-intensive. While we agree that advanced LLMs with carefully crafted prompts could potentially detect such attacks, we note that the same advancements in LLM capabilities and prompt engineering could also improve the stealth of our poisoning via SC. This creates an evolving dynamic in which both attack and defense methods can benefit from progress in LLMs.
> > > >
> > > > We sincerely appreciate your constructive reviews of our submission and are grateful for your reconsideration of the score. Thank you again for your time and valuable insights :)

---

### Decision · Program_Chairs · 2025-09-17

**Decision:**

Accept (spotlight)

**Comment:**

The key research topic in this paper is the propagation of data poisoning attacks through the generation and use of synthetic data. The authors discuss why the traditional data poisoning methods are ineffective in this scenario -- a distributional mismatch between the narrow poisoning data and the broad queries used for synthetic data generation. To overcome this limitation, the paper introduces the Virus Infection Attack (VIA), a universal framework designed to make existing poisoning attacks contagious through synthetic data. In terms of strength, the threat model is both novel and highly relevant, as noted by all reviewers. The proposed VIA method is conceptually straightforward and directly addresses the core reason why standard poisoning fails to propagate. Taking the existing paper and new results from the rebuttal together, authors have addressed the main weaknesses of the initial submission. Reviewer Xcj8 and Reviewer 7ogA explicitly stating they were raising their scores as a result because the concerns in evaluations are address. Thus, I believe this paper is ready for publication in this venue and the underlying issue should be more broadly discusses with the academic audience.